# Education for non-citizen children in Malaysia during the COVID-19 pandemic: A qualitative study

Tharani Loganathan[1]*, Zhie X. Chan[1], Fikri Hassan[1], Watinee Kunpeuk[2], Rapeepong Suphanchaimat[2,3], Huso Yi[4], Hazreen Abdul Majid[5,6]

1 Centre for Epidemiology and Evidence-Based Practice, Department of Social and Preventive Medicine, University of Malaya, Kuala Lumpur, Malaysia, 2 International Health Policy Program, Ministry of Public Health, Nonthaburi, Thailand, 3 Division of Epidemiology, Department of Disease Control, Ministry of Public Health, Nonthaburi, Thailand, 4 Saw Swee Hock School of Public Health, National University of Singapore and National University Health System, Singapore, Singapore, 5 Centre for Population Health, Department of Social and Preventive Medicine, University of Malaya, Kuala Lumpur, Malaysia, 6 Department of Nutrition, Faculty of Public Health, Universitas Airlangga, Surabaya, Indonesia

* drtharani@ummc.edu.my

**Data Availability Statement:** All relevant data are within the manuscript and its Supporting Information files.

## Abstract

The COVID-19 pandemic disrupted schooling for children worldwide. Most vulnerable are non-citizen children without access to public education. This study aims to explore challenges faced in achieving education access for children of refugee and asylum-seekers, migrant workers, stateless and undocumented persons in Malaysia during the pandemic. In-depth interviews of 33 stakeholders were conducted from June 2020 to March 2021. Data were thematically analysed. Our findings suggest that lockdowns disproportionately impacted non-citizen households as employment, food and housing insecurity were compounded by xenophobia, exacerbating pre-existing inequities. School closures disrupted school meals and deprived children of social interaction needed for mental wellbeing. Many non-citizen children were unable to participate in online learning due to the scarcity of digital devices, and poor internet connectivity, parental support, and home learning environments. Teachers were forced to adapt to online learning and adopt alternative arrangements to ensure continuity of learning and prevent school dropouts. The lack of government oversight over learning centres meant that measures taken were not uniform. The COVID-19 pandemic presents an opportunity for the design of more inclusive national educational policies, by recognising and supporting informal learning centres, to ensure that no child is left behind.

## Introduction

The coronavirus disease 2019 (COVID-19) pandemic has disrupted schooling for billions of children worldwide [1], with countries implementing school closures to reduce transmission of the severe acute respiratory syndrome coronavirus 2 (SARS-CoV-2). Most vulnerable are

**Funding:** This research was funded by The Asia Pacific Observatory (APO) on Health Systems and Policies grant number IF034-2020 awarded to Tharani Loganathan. The funders had no role in study design, data collection and analysis, decision to publish, or preparation of the manuscript.

**Competing interests:** The authors have declared that no competing interests exist.

marginalised children, including non-citizens with pre-existing disparities in access to education [2].

In Malaysia, several groups of non-citizen children are considered marginalised including refugees and asylum-seekers in Peninsular Malaysia, children of migrant workers, stateless and undocumented persons in the state of Sabah in East Malaysia. Due to the non-ratification of the 1951 Convention on the Status of Refugees and its 1967 Protocol, and the 1954 Convention on the Status of Stateless Persons [3–5], Malaysian immigration law does not distinguish refugees, asylum-seekers, or stateless persons from undocumented migrants. As such they have limited entitlements to education, healthcare, social protection or formal employment, and are at risk of being arrested and detained for immigration offences [6–8].

Malaysia is host to at least 3.2 million non-citizens or 10 per cent of its population in 2019 [9]. More than 178,000 refugees and asylum-seekers are registered with the UNHCR in Malaysia, including 46,000 children under 18 years. While the majority are stateless Rohingya (102,000) and other minorities (52,000) from Myanmar (154,000), Malaysia hosts refugees from some 50 countries including Pakistan, Yemen, Syria, Somalia, Afghanistan, Sri Lanka, Iraq, and Palestine. Refugees live in urban, non-camp settings scattered around Peninsular Malaysia, with most concentrated around the capital city of Kuala Lumpur [10], and the surrounding Klang Valley.

Estimating the number of stateless and undocumented children in Malaysia is difficult, especially in Sabah, where cross-border migration with the neighbouring Philippines and Indonesia is common. Nevertheless, 290,347 stateless children under 18 years old in Malaysia were reported in 2016 [11]. Sabah hosts diverse non-citizen populations, including stateless persons from the seminomadic Bajau Laut community, undocumented indigenous peoples, undocumented Filipino and Indonesian migrants, and documented Indonesian plantation workers [12–15].

The right of every child to education on the basis of equal opportunity is recognised in the Convention of the Rights of the Child (CRC), of which Malaysia is a signatory. However, reservations made particularly to article 28 paragraph 1(a) demonstrates that commitments made towards education have not been expanded beyond citizenship [16, 17]. This is compounded by the 2002 amendment to the Education Act 1996 (Act 550), which made primary education compulsory for all Malaysian children in exclusion of non-citizens [18].

Unable to enter public schools, non-citizen children in Malaysia have limited access to education [19]. They are reliant on the informal education system not funded or regulated by the state, in the form of alternative or community learning centres (LCs) supported by civil society organisations, faith-based organisations, private donors and local communities [20, 21].

In Malaysia, schools first closed when the Movement Control Order (MCO) was declared on 18th March 2020 in response to the COVID-19 pandemic [22, 23] (See Fig 1). Repeated school closures and movement restrictions have since disrupted formal lessons, with students and teachers forced to adapt to new learning modalities like online learning, seen as the best alternative to ensure the continuity of learning [24]. In this study, we explore perspectives from stakeholders to understand the experiences and challenges faced by marginalised non-citizens and those supporting them, in accessing education during the pandemic.

## Materials and methods

### Overview

We used qualitative methods to explore experiences and challenges faced by marginalised non-citizen children in Malaysia and those supporting them, in accessing education during the COVID-19 pandemic. This study focuses on refugee and asylum seeker, migrant, stateless

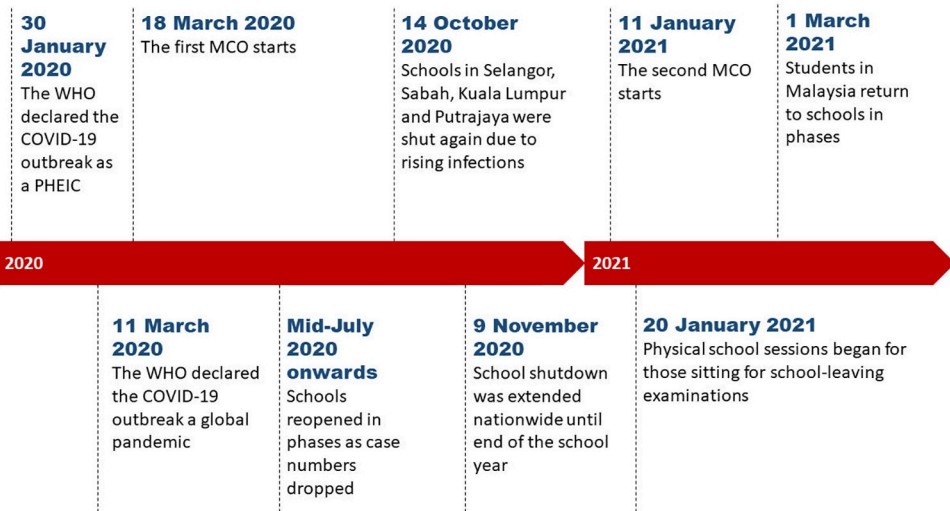

**Fig 1. Timeline of school closures and reopenings in Malaysia during the COVID-19 pandemic from January 2020 until March 2021.** WHO—World Health Organization; MCO—Movement Control Order; PHEIC—Public Health Emergency of International Concern. Sourced from [22, 23, 25].

and undocumented children in Malaysia. International students and children of expatriates were excluded from this study. (Refer to S1 File for the definition of terms)

## Recruitment

Participants were recruited to represent the diversity of experiences of different non-citizen groups in Malaysia. We purposefully recruited participants from Linked-In and other sites, according to their professional backgrounds and experience with non-citizen groups. Additional participants were recruited via snowball sampling of interviewees, until researchers agreed that further interviews would not yield additional information, as thematic saturation was reached. Potential participants were invited to participate by telephone and emails and were sent participant information sheets and consent forms.

## Data collection

Data collection was conducted from June 2020 to March 2021. We conducted 32 in-depth interviews, interviewing 33 individuals. Most interviews were conducted on an individual basis; however, two interviews were conducted with 2 participants from the same organisation. One participant was interviewed twice.

Due to physical distancing requirements and precautionary measures taken during the MCO periods, most interviews (28) were conducted remotely using online video conferencing tools (Microsoft Teams and Zoom), while 5 interviews were conducted in-person in Kuala Lumpur, Malaysia.

Study participants were community organisers from civil society organisations, education providers from learning centres, policymakers from government and international organisations, and researchers with professional expertise in education of refugee, stateless and migrant populations in Malaysia. We interviewed adult refugees, who were former students with experience navigating the education system as children. Several participants interviewed had specific expertise with certain non-citizen groups: the stateless and undocumented persons in

**Table 1. Characteristics of the study participants (n = 33).**

| Respondents' primary role | | Label | No. |
|---|---|---|---|
| Community organiser | | CO | 4 |
| Former students[1] | | FS | 7 |
| Education provider[2] | | EP | 11 |
| Policymaker | | POL | 4 |
| Researcher | | RES | 7 |
| **Total** | | | **33** |
| **Respondent's citizenship status** | | **Female** | **Male** |
| Malaysian citizen | | 15 | 7 |
| Non-Malaysian citizen | | 8 | 3 |
| **Total** | | **23** | **10** |
| **Non-citizen type** | **Peninsular Malaysia** | **Sabah** | **Overall Malaysia** |
| Overall—non-citizens | | | 5 |
| Refugees | 15 | | |
| Stateless | 3 | 7 | 2 |
| Migrant | | | 1 |
| **Total** | **18** | **7** | **8** |

[1]All the former students interviewed were adult refugees. Of the 7 interviewed, 3 were also education providers.

[2] Of the 11 education providers interviewed, 7 also identified themselves as community organisers.

Sabah (7), and refugees in Peninsular Malaysia (18). Sample characteristics are shown in Table 1.

Semi-structured interview guides were developed to seek participants' perspective on education access for non-citizen children in Malaysia during the COVID-19 pandemic. See S1 File for interview guides. Interviews averaged from 1 to 1.5 hours in length and were conducted either in English or Bahasa Malaysia (Malay language) depending on the participants' preference. Audio recordings were transcribed verbatim. Concurrent analysis informed the further refinement of question guides.

## Data analysis

Data analysis was conducted in an immersive, exploratory, and inductive manner, with regular discussions between researchers to refine codes and identify new themes. Thematic analysis was conducted as described by Braun and Clarke, where themes were identified and reported using six phases: (1) becoming familiar with the data, (2) generating initial codes, (3) searching for themes, (4) reviewing themes, (5) defining themes, and (6) producing the report [26]. Transcripts were coded into emerging themes using NVivo 12 Plus, (QSR International, Melbourne, Australia) and quotations were extracted into Microsoft® Excel® for Office 365, (Microsoft, Redmond, WA, USA). Interviews in Bahasa Malaysia were analysed in the same language, while extracted quotations were translated to English for publication.

## Ethics

Participant information sheets distributed detailed participants rights, the potential risks and benefits from the study, confidentiality, and study procedures including audio recording and usage of data. Verbal and written informed consent were obtained before commencing interviews. Participants were informed that study participation was entirely voluntary, and they were free to refuse to answer questions or terminate interviews at any point. All participants

agreed to be audio recorded and quoted anonymously in publications. Audio recordings and electronic transcripts were stored in secure data servers.

This study was approved by the Medical Ethics Committee, University Malaya Medical Center (Reference: UM.TNC 2/UMREC).

## Results

Study findings are presented by major emerging themes: COVID-19 impact on non-citizens communities, challenges with online learning, measures to ensure continuity of learning and closure of learning centres and non-educational contributions. Table 2 summaries the main study findings.

### COVID-19 impact on non-citizen communities

**Loss of livelihoods and competing priorities.** During the COVID-19 pandemic and subsequent movement control orders (MCOs), communities faced job loss, as they were unable to continue working in the informal sector. Many were unable to afford rent and faced evictions.

**Table 2. Main findings of the study.**

**COVID-19 impact on non-citizen communities**

- Education not prioritised as the community faced job loss, evictions and others
- Xenophobia and scapegoating during the pandemic exacerbated vulnerabilities
- Children drop out from school to help financially support the family

**Challenges with online learning**

- Lack of digital devices, data and internet connectivity among non-citizen children
- Digital divide more prominent in rural areas and East Malaysia
- Teachers forced into online teaching with little preparation
- Teachers unfamiliar with the use of digital devices and the pedagogy of online teaching
- Online learning unable to replace face-to-face interaction with students
- Lack of accountability among students when participating with lessons
- Support and guidance from caregivers necessary for online learning
- Illiterate parents are unable to guide children with home-based learning
- Living environment not conducive for home-based learning
- Lack of peer support and language skills hamper learning for younger students
- Mental health unaddressed as students are deprived of support from peers and teachers

**Measures to ensure continuity of learning**

- Teachers gradually adapted to online teaching with training and guidance
- Learning centres with internet connectivity and prior experience adapted rapidly
- Learning centres offer flexible schedules to encourage students to stay in school
- Innovative national education platforms have potential to include non-citizen children
- Alternative education measures implemented when online education was impossible
- Distribution of learning material was often done along with the delivery of food aid
- Small group classes conducted at homes in rural setting with poor internet connectivity
- Students sitting for school leaving examinations prioritised with lessons and resources

**Closure of learning centres and non-educational measures**

- Smaller learning centres shut down due to loss of revenues from school fees
- Donors prioritised food donations rather than funding learning centres initially
- Fundraisers later allowed for the purchase of devices to enable distance learning
- Some learning centres discounted fees to enable students to continue schooling
- Other centres were unable to provide discounts as they struggled to transition online

This interviewee explained that education was not prioritised when livelihoods are threatened, as there were other competing priorities faced during the pandemic.

*"[COVID-19] The impact on the schools? Okay, number one, schools [were closed], number two, many of the parents are unemployed. You know why? The new minister of XXX? He doesn't allow foreign workers to work in the wet market and that is their main job. [And] there are a lot of them. So, the other day, the school administrator said, 'What is the use of schools, [when] they were thrown out of their houses?* RES-03 (September 2020) partially translated from the Malay language.

Participants explained that many refugee families were poor due to their inability to work legally in Malaysia. The pandemic and subsequent control measures forced refugee families into further impoverishment.

**Immigration raids and xenophobia.** Beginning May 1st, 2020, immigration raids were conducted to detain undocumented migrants found in areas under lock-down for COVID-19. The Malaysian government's hard-line stance to detain and deport non-citizens without valid documents during a public health emergency was a major concern to the sizable population of undocumented migrants in the country. This participant shared that the ever-present fear of arrest experienced by non-citizens was heightened during the pandemic; disrupting non-citizen students' access to education.

*"What was fragmented before [education access], it is almost non-existent now. Because there was the danger; not just a physical danger of COVID, like a health pandemic danger so you had to kind of keep away. But also, with the whole persecution of undocumented non-citizen children being hauled up on [immigration] trucks! I knew of communities who didn't leave their house at all, not even to get food, because of that fear of persecution. The fear has always been there."* POL-02 (July 2020)

Interviewees also brought up increased xenophobia, as foreigners were blamed for bringing in the virus from neighbouring countries with higher case numbers. Non-citizens were also perceived as likely to be infectious. This sentiment was particularly prevalent in Sabah, which shares borders with the Philippines and Indonesia.

*"The locals blame the Filipinos and Indonesians for bringing in disease. And yes, so they are trying to get rid of them and send them all back. But send them back to where? Their home is here. They do not have a home outside of Sabah."* EP-04 (August 2020)

This participant alluded to the complex migration situation in Sabah, explaining that many stateless communities have been in the country for generations.

**Children drop out of school to support families.** Since there are no legal provisions to mandate compulsory education for non-citizen children, parents have the sole responsibility of ensuring their children complete their education. Some refugee parents showed reluctance to invest in the education of their children as it is hard to imagine a future in Malaysia. This is indicative of the lack of opportunities available to non-citizens to pursue tertiary education and obtain formal employment. Study participants were concerned that pandemic hardships may have facilitated children dropping out of schools to support their families.

*"Some students who have dropped out because the parents said that 'No, they need to work, we don't have an income. The father has lost a job. The mother has lost the job.' I know they cannot work legally, but illegally they do work."* EP-03 (August 2020)

This interviewee also explained that older siblings are often seen as responsible for contributing to the household income and may be forced to drop out of school early.

## Challenges with online learning

**Devices, data, and connectivity.**   The nationwide school closures forced the rapid adoption of online learning. Study participants agreed that smartphones are considered as luxury goods across marginalised population groups irrespective of citizenship status. While the option for online learning was made available to some children, most non-citizen families encounter limited access to digital devices and internet connectivity.

*"They might not have access even to mobile top-ups. They might not have enough resources even to [buy] tablets or mobile phones. So, this is super, super challenging. I do know that communities are really struggling because students cannot just continue education, because they don't have devices [and] they don't have internet. And the parents are like: 'Okay. What can we do? We can't do anything! We have to bring our mobile to work as well. . . We need the mobile phone!' If they leave it at home for you to study and they [the parents] will not be able to do their work."* FS-05 (March 2021)

Interviewees described unhelpful family situations where a smartphone was shared by several siblings, and this resulted in poorly attended classes and unsubmitted homework.

The digital divide is even more pronounced in the interior regions of Sabah and Sarawak in East Malaysia, where poor telecommunications infrastructure had resulted in limited internet connectivity.

*"In urban areas, perhaps internet access is still not a big problem. If they cannot afford it, I would say that we are very lucky that some of the telco [telecommunication] companies are providing free data packages for the students. But in rural areas, especially in Sabah and Sarawak [East Malaysia], we have places whereby internet access is a luxury. We have areas whereby the parents simply could not afford any tablets. And so, for them [the teachers] to conduct online teaching and for [the students] to continue with home-based learning–it would be very, very challenging."* POL-01 (July 2021)

Participants informed that it was nearly impossible for children living in rural areas with limited internet connectivity to participate in online learning.

**Teachers struggle with online learning.**   Teachers were forced to rapidly adapt to the uncertainties brought upon by the pandemic, movement restrictions, and school closures. Interviewees explained that many teachers were unprepared for the sudden shift to online teaching as they were unfamiliar with the use of technology and were only trained in the conventional mode of teaching.

*"I think it was just that when the MCO started and the lockdown happened, it was the most unpredictable times for teachers. Teachers don't know what to do during these times. Everything was so unpredictable, and we were not prepared at all! I think nobody was prepared for this technology [adoption]. Everything pretty much shifted online and everybody was anticipating that maybe it would last for a few months and then the vaccine will be here, and things*

*will be better. But then it went on for full last year. And this year as well, the same thing is going on. So, it has been so unpredictable, so difficult and so challenging, because as teachers, what we are trained to do is the conventional teaching style–[to interact] physically with the students."* FS-05 (March 2021)

Participants felt that technology should be a supplementary tool to enhance the teaching learning environment, and not as a replacement for a teacher or the physical learning environment. The physical classroom provides face-to-face interaction with students; within a controlled environment away from external distractions. This participant explained that remote learning does not foster student accountability in terms of participation in lessons, homework, and examinations.

*"It is just so, so difficult online. Definitely, [with] online [teaching], you can't be forcing somebody to reply. The accountability part is also not there from the students' point of view. For example, if they are given an online exam—how much time you can set? What measures can you take? You still don't know if they are cheating behind the screen or if they are 'googling' their answers and all that!"* FS-05 (March 2021)

Teachers found it difficult to monitor every student and often relied on parents or guardians to facilitate effective learning. Participants felt that students without parental support or teachers to check in on them were more likely to drop out from online classes.

**Student struggle with online learning.**   Study participants informed that living conditions during the COVID-19 pandemic were not conducive for remote learning. Refugees living in urban areas often live in cramped environments that pose a challenge to remote learning. Forced eviction further exacerbate the issue of the lack of space and privacy for effective learning.

*"I realised that they [students] needed people to call and chase them, if not they wouldn't do it [the homework]. Not because they didn't want to do it. But because of the situation at home. Now, you must understand that some refugee families live in big, extended families. And because they cannot afford rental during COVID-19. So, they were thrown out, evicted from their house. And they had to go and stay with like another friend or another family and all that. So, at one stage there was one family that came to us and reported that there was like 3 families living in a three-room apartment, each family got 6 to 7 members. Okay, how do you study in this situation? How do you do your homework in this situation?"* EP-03 (August 2020)

Participants explained that most students were unable to have private space to study or attend lessons without distractions from family members.

Teachers interviewed shared difficulties instructing younger students and those with poor language skills. In normal settings, these students would have had extra guidance from teachers, as well as peer support to help keep up with lessons.

*"And often, like for Rohingyas for example, because the majority of the teachers are local and children who join the class late rely on peer support to help translate and to help them catch up with the Bahasa [Malay language]. So, when they are isolated at home and in quarantine, they are not able to even communicate with their friends to ask for help."* POL-03 (August 2020)

In addition, participants shared that younger students faced more disparity as they were less likely to have a digital device.

Study participants explained that mental health issues were a concern for refugee communities and their children during the pandemic. In addition to the anxiety related to the COVID-19 pandemic, children experienced stress from the inability to fully participate in online learning. This interviewee explained that the closure of learning centres may mean the loss of support systems, which may impact a child's mental health.

*"Our biggest problem is we don't address our mental health. By 'ours' I meant the refugee community. When COVID-19 [struck], the children are unable to go to school, where they [would] spend their days. Even though it is not a very quality education system, but still, they have friends to talk to. They feel relief from all the daily stress, where they have to see their mother and father struggling. Because we tend to give that negativity to our [friends], the persons who love us [and] actually want to come and talk to us. Refugee children hardly speak to their parents, because they are fearful, and they don't know if it's appropriate to tell the parents that the day was bad."* FS-01 (August 2020)

**Parental support necessary to facilitate learning.** This interviewee explained that especially for younger children, it is often necessary for a parent or guardian to participate in online learning.

*"When the father and mother are working, then nobody is there to take care of the child. Even if somebody is there to take care of the child. The person can just take care, but they cannot just run online learning at home, together with the teacher. So, it is not just difficult for teachers to adapt to this environment. I think students are also super, super affected by this."* FS-05 (March 2021)

Some parents were able to take up the mantel of teaching to the benefit of their children. However, this interviewee explains that illiterate parents may not be able to assist in this way.

*"Refugee children lost almost one year of education because of COVID-19. Some parents, parents who are well educated and parents who take education very seriously have been home tutoring their children since the COVID-19 [lockdown measures]. Like you know, teaching them using the books and syllabus that they already have. But some parents who are illiterate themselves, they can't do much for their children. So, their children are basically just spending the year doing nothing at home. Just playing and having fun and not actually learning anything because they can't go to school and they can't access online learning as well."* FS-03 (November 2020)

Participants shared that unlike the average Malaysian parents who are literate and have basic knowledge of using technology, parents from marginalised non-citizen communities like the Rohingya have limited literacy and were unable to help their children manoeuvre the challenges of online learning.

## Measures to ensure the continuity of education

**Adaptive measures for online learning.** During the pandemic, learning centres had limited resources and many had ceased operations. Others, however, appealed to sponsors and funders to finance remote learning and provide devices and data packages for students. After a

period of transition, teachers eventually adapted to online teaching with experience, guidance, and appropriate training.

This policymaker informed that in urban areas where internet connectivity was not an issue, teachers adapted well and there was even active creation of material for online learning.

*"In plantations, they have some difficulty in CLCs [community learning centres], because not all plantations have internet access. For some, it is okay to use online learning because we provide material in online learning–for X and Y [learning centres in urban areas] it is easy for them. And this COVID-19 triggers them to [become more] innovative. They create many new materials for online learning. For 3 to 4 months only they are trying this online method. It is okay."* POL-04 (October 2020)

Educationists informed that some forward-thinking providers had digital learning in place before school closures and were able to rapidly adapt to remote learning.

Some learning centres made changes in class schedules and content in an attempt to keep students in school. This participant shared examples of flexible class schedules offered to enable working students to continue with their lessons, as well as innovative skills training to promote employability and alternative income sources.

*"We are operating the school now in two sessions. We have a morning session and an afternoon session. So, you can choose if you have some way of earning part-time money. You can choose to either work in the morning and come to school in the afternoon or come to school in the morning and work in the afternoon. We are trying to be as flexible as possible to keep the child in school. We are also trying other options like teaching them to earn [money] online."* EP-03 (August 2020)

The pandemic also forced the Malaysian government to create innovative education platforms for remote teaching, including digital content and television broadcasts. Some participants saw this as an opportunity to accelerate access to education as these platforms could be extended to non-citizen populations.

*"You see, you already have a digital platform with specific login [details] and COVID actually, for me, has been a great opportunity to accelerate things, so whatever you are doing, you say, 'Okay, let's look at how we can [do things better].' So, we then say that 'Can we look at extending this to the non-Malaysian population?'"* POL-02 (July 2020)

Others felt that innovations were impeded by slow implementation and the lack of inclusivity in the language of content which are concerns in multicultural Malaysia.

**Alternative arrangements.**  Education providers interviewed expressed doubt on the usefulness of online learning in the long term. Participants felt that technology should be a supplementary tool to enhance teaching and not as a replacement for a teacher and the physical learning environment.

This participant shared that especially in rural areas where poor internet connectivity made online learning impossible, teachers provided physical materials for students to study at home.

*"Schooling stopped and the teacher would provide [students with] materials in hard copy if the internet is not available. [They would] provide to the parents or students [materials to study at home] without any face-to-face class lah."* POL-04 (October 2020)

Learning materials were prepared beforehand and readied for pick up and drop off upon completion of homework. Participants shared that the distribution of learning material was often done together with the delivery of food aid.

*"There were partners that distributed out learning materials and homework with the food assistance packages. So, they had a system where they provide the students' families with food aid. Students are supposed to do their homework and return it to the teachers in a week or two weeks when the learning centre's partner goes again with another round of food assistance."* POL-03 (August 2020)

Participants informed that in some cases, teachers would go from house to house to teach students in smaller groups to help address barriers of remote learning. This was a practical solution for the lack of internet connectivity and devices in rural areas.

*"[We were] supposed to be learning online. Well, of course, schools do not have computers. They do not have access to the Internet. There is no [internet] line whatsoever in interior regions. But that problem was solved because the teachers went house to house. So, they just go around to each of the students in the school and teach them in their house, yeah. They might gather a few neighbours. I know you are not supposed to meet, but in very rural areas where there was no COVID-19, nobody minds and they would get maybe two families together, maybe 5 to 6 kids, and then teach them. And then the next day they go to another house [in sessions of]- morning, afternoon, and night. They were even teaching them at night!"* EP-04 (August 2020)

Educationists shared that teaching smaller groups was especially helpful for beginners and less IT savvy students that need more attention. One participant shared that small group sessions had an unexpected benefit of increased female participation, as parents' security concerns were addressed.

*"There were also teachers that were conducting what we call 'cluster classrooms', where there is a teacher that goes to the students and [will] have discussions on a rotation basis with small groups. While they were doing this, they discovered that there were girls who were previously not allowed to go to school. Suddenly [these girls were] wanting to participate and were allowed to participate. The reason being, some of these parents understand the need for education, but they are just so concerned about safety and security. So, when education is brought to the community, at their homes, then the parents are okay with the girls participating."* POL-03 (August 2020)

Study participants informed that home classes were only conducted when case numbers were relatively low before October 2020. Educationists interviewed in early 2021, described physical visits to students' home as not possible at that time because of increased risk of COVID-19 transmission and the legal repercussions of not complying with movement restrictions.

**School leaving examinations prioritised.**   Educationists informed that learning centres prioritised selected students who would be sitting for IGCSE (International General Certificate of Secondary Education) exams.

*"For our IGCSE students, actually when the MCO started in March we were frantic and panicked. But because the number of students [are] very small, only eight of them. So, we bought*

*data for every one of them. We bought SIM cards and then we allow them to take back our school laptop."* EP-08 (December 2020)

As only a few learning centres prepare students for IGCSE examinations, these students are considered important investments and limited resources like laptops and data packages for internet connectivity were allocated for them.

## Closure of learning centres and non-educational contributions

**Impact of closures on learning centres.** The lockdowns and the forced adoption of online teaching saw smaller learning centres forced to suspend operations due to limited resources. Many learning centres were unable to collect school fees, thus were unable to pay rental or teachers' salaries.

*"During the MCO so many of the learning centres were so badly affected because they were not able to collect any school fees and thus weren't able to pay the teachers anything or pay for rental [of school premises]."* POL-03 (August 2020).

Participants explained that especially during the first MCO, donors prioritised food aid as an immediate concern and pulled out from funding learning centres. This left many learning centres unable to provide students with devices and data for remote learning.

**Food aid and non-educational measures.** School meals programme is an essential component of support provided at learning centres in Malaysia. Non-citizen families are described by participants as being food insecure, with some children having only one cooked meal per day. During the pandemic, school closure interrupted regular meals provided by school meals programmes, while families struggled with interrupted livelihood. Study participants explained that food aid organised by civil society organisations were vital during the lockdown.

*"Because parents were not earning much, so they [children] don't have many meals a day, they usually have one meal a day. And we have managed to address this issue by getting funding to cover some food to be supplied in school. School meals specifically to the younger children. We provide them with cereal, we provide them with milk. Now we are giving hot meals to about, not all students, maybe 60 to 70 students who don't get meals at home."* EP-03 (August 2020)

Participants shared that fundraising efforts conducted by civil society organisations were supported by the Malaysian public and enabled the provision of food aid and other necessities to assist non-citizens communities during the pandemic. As the pandemic progressed, funds were raised to allow the purchase of devices, teacher training and others to enable learning centres to conduct remote teaching.

**School fees and funding learning centres.** Interviewees shared that even though school fees were low, they were still a hurdle for marginalised populations. Non-citizen families were faced with financial difficulties due to job loss and immigration crackdowns against undocumented migrants. Educationists shared that some learning centres discounted tuition fees, as their main aim was to enable children to continue their education and not to generate profit.

*"Preschool [school fees] in Klang Valley is RM 50, Primary [school fees]is RM70, Secondary [school fees] is RM120. That was pre-COVID-19 [fees]. Post COVID-19 [fees]-Yeah, we have cut down our fees for Preschool and Primary to just RM10 and Secondary to RM50, because we want the parents to bring their children back to school, as a priority lah."* EP-08 (December 2020)

However, other education providers expressed that the lack of funding hampered their efforts to properly migrate to online learning. This interviewee shared the contrasting perspectives expressed by parents and education providers, where parents expected fee reductions undervaluing education providers struggle to support the transition to online education.

*"You know when the lockdown happened, and everything moved to online learning, the schools were trying to support students in many different ways. The worst thing was that parents have these thinking, 'Oh, online learning is happening. There should be a reduction in school fees! Because we are not using the facilities and all!' In a way it is okay, you know if you look at it from both [parents and schools] perspectives, as a parent, you would say, 'I would like to have a reduction [in school fees]!' But as somebody who is running the school, you will be like, 'Oh, I am running out of resources! I need to ensure that all my kids who are studying in my school that they have devices and Internet access! You know, and they have no digital devices!"* FS-05

## Discussion

Unplanned school closures cause distress to students, teachers and parents while being detrimental to a child's educational achievement, social development, physical and mental health [27–32]. School closures disrupt essential school meals, social interaction and support from peers and teachers essential to a child's physical and mental wellbeing, while exposing children to potentially unhealthy home environments [33, 34]. We found that lockdowns are particularly troubling to underprivileged non-citizen households in Malaysia as employment, food and housing insecurity are compounded by increasing xenophobia and scapegoating of migrants [35–38], exacerbating pre-existing economic and social inequalities [39, 40].

Online learning is seen as essential to ensure continuity in education. However, the rapid adoption of technology in the face of crisis may widen inequalities faced by low-income, rural, and marginalised populations [2, 24, 41–44]. Our findings suggest that non-citizen children may lose out on learning opportunities due to a lack of stable internet connections, data and digital devices like computers, tablets, and mobile phones, which are seen as luxury goods. Poor telecommunications infrastructure in rural and remote regions exacerbates digital inequalities as internet connectivity is a prerequisite to online learning [45, 46]. According to the Department of Statistics Malaysia, household internet penetration in Malaysia increased from 87% in 2018 to 90.1% in 2019, and yet there remains a disparity in internet connectivity with East Malaysia being underserved [47–49]. Poor connectivity and lack of devices [50, 51] may influence the uptake and acceptance of online learning, as reflected by findings of a nationwide survey, in which students from East Malaysia indicated higher levels of negative feelings towards online learning [52].

Other than access to technology, digital literacy is crucial for navigating online learning and is much dependent on the skills of students, teachers, and parents [41, 45, 53]. We found that younger students, particularly those with poor language skills have difficulty following virtual lessons and would benefit from parental guidance in the absence of a teacher. Unfortunately, non-citizen parents with low educational attainment, like the Rohingya refugees in Malaysia [54, 55], may be less able to support children with home education and online learning [43, 56–58]. Studies elsewhere have linked various dimensions of socioeconomic background, especially social housing and parental education with school absenteeism and poor academic achievement [59–61].

Additionally, the home learning environment is an important factor influencing home-based learning [41, 58]. This includes the availability of a quiet room for schoolwork, which is likely lacking in the cramped households of disadvantaged non-citizen families. Previous studies indicate that refugee families tend to share living spaces to minimise household expenses [62, 63], and living conditions may have worsened by the economic crisis and social upheavals experienced with the pandemic [64, 65].

Teachers were forced to rapidly transition to online teaching and faced numerous challenges including poor online teaching infrastructure, lack of digital skills and relative inexperience. Our findings suggest that teachers habituated to conventional teaching practices may be less familiar with the pedagogy of online instruction. Training and guidance are therefore essential for teachers to enable them to best incorporate technology into online instruction [56, 66, 67].

Even before the COVID-19 crisis, studies have shown that non-citizen children in Sabah were more likely to be out-of-school, at all levels of schooling [68]. Also, dropout rates increase as refugee youth in Peninsular Malaysia approach adolescence, joining the workforce primarily out of economic necessity [62]. Unfortunately, parental job loss coupled with school closures during the pandemic may have hastened children's entry into the labour market [33, 41], while unaffordable school fees may make the return to school difficult.

As online learning was near impossible for many students, we found that alternative arrangements for home-based learning using low-tech or no-tech solutions were developed based on necessity. Such innovative measures were crucial in ensuring continuity in education, particularly among marginalised groups unlikely to participate in online learning [69]. And yet, our findings suggest that the lack of government oversight over learning centres meant that measures taken to ensure continuity of education were not uniform. In Thailand, despite Thai public schools being free for all children, alternative education in the form of migrant learning centres (MLCs) was preferable to migrant children, as culturally sensitive services were provided. Nonetheless, since MLCs are informal, the quality of education offered was highly inconsistent due to the lack of policy regulation, in addition to budgetary and human resource constraints [70].

Though there is no straightforward solution to ensuring rights to education access for vulnerable population, this study can serve as a useful input for future policy design to protect education rights for vulnerable children in the time of crisis. First, policy makers should be aware that the one-size-fit-all measure does not always lead to favourable outcomes. The education system for vulnerable children, particularly the non-citizens, in many places differ greatly from the mainstream education for the citizens. The abrupt change in policies (such as implanting an on-line learning or school closure), if done without supporting systems in place (in terms of budget, technology, finance and know-how), may cause longterm damage on the children (as a result of school dropout or poor learning outcomes) than the shortterm health damage caused by COVID-19. Preparedness plan for the education system for any future crises should seriously account for the diverse nature of education system for non-citizen children.

This study has several limitations. Firstly, we chose not to interview children under the age of 18 years to avoid ethical challenges. Instead, we interviewed adult non-citizens, as well as representatives of learning centres and others able to share community experience on education during the pandemic. Secondly, the qualitative nature of this study does not allow the generalisation of findings. Nevertheless, we hoped to have captured a diversity of experiences by interviewing stakeholders from different non-citizen communities throughout Malaysia, including the state of Sabah, which hosts large populations of stateless persons, migrants, and refugees.

This study has several strengths. This study provides a unique case study of an informal education system catering to marginalised non-citizens in an upper-middle-income country during the COVID-19 pandemic. This work will provide a vital understanding of some of the difficulties faced by vulnerable children during the pandemic and identifies entry points for policy intervention. Future research should explore the consequences of school closures on marginalised children including long-term learning loss and health impacts such as poor nutrition and mental health.

## Conclusion

The COVID-19 pandemic and measures taken to combat it have exacerbated pre-existing vulnerabilities faced by marginalised non-citizens. Deprivation of schooling is especially harmful to children from poor socioeconomic backgrounds, as education is an enabling right essential for individuals, families, and communities to escape poverty.

The COVID-19 crisis is an opportunity for governments to do better and design inclusive, equitable and quality education policies that are linguistically and culturally sensitive. We recommend that the Malaysian government takes an active responsibility in ensuring the Right to Education for all children in the country, by recognising and supporting informal learning centres.

## Supporting information

**S1 File. Interview guide.**
(PDF)

## Author Contributions

**Conceptualization:** Tharani Loganathan, Zhie X. Chan, Fikri Hassan, Hazreen Abdul Majid.

**Data curation:** Tharani Loganathan, Zhie X. Chan, Fikri Hassan.

**Formal analysis:** Tharani Loganathan, Zhie X. Chan, Fikri Hassan.

**Funding acquisition:** Tharani Loganathan, Watinee Kunpeuk, Rapeepong Suphanchaimat.

**Investigation:** Tharani Loganathan, Zhie X. Chan.

**Methodology:** Tharani Loganathan, Fikri Hassan.

**Project administration:** Tharani Loganathan, Zhie X. Chan, Watinee Kunpeuk.

**Resources:** Tharani Loganathan, Fikri Hassan.

**Software:** Tharani Loganathan, Zhie X. Chan.

**Supervision:** Tharani Loganathan, Hazreen Abdul Majid.

**Visualization:** Tharani Loganathan, Zhie X. Chan, Hazreen Abdul Majid.

**Writing – original draft:** Tharani Loganathan.

**Writing – review & editing:** Tharani Loganathan, Zhie X. Chan, Fikri Hassan, Watinee Kunpeuk, Rapeepong Suphanchaimat, Huso Yi, Hazreen Abdul Majid.

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
