## [Decision Letter · Decision Letter 0]

21 Sep 2021

PONE-D-21-23915Education for non-citizen children in Malaysia during the COVID-19 pandemic: A qualitative studyPLOS ONE

Dear Dr.  Loganathan,

Thank you for submitting your manuscript to PLOS ONE. After careful consideration, we feel that it has merit but does not fully meet PLOS ONE’s publication criteria as it currently stands. Therefore, we invite you to submit a revised version of the manuscript that addresses the points raised during the review process.

We look forward to receiving your revised manuscript.

Kind regards,

Maryam Farooqui, Ph.D

Academic Editor

PLOS ONE

Journal Requirements:

"Funding was from National Science Foundation under Grant Number 2051510 to Carlos F Martino"

"Funding was from National Science Foundation under Grant Number 2051510

to Carlos F Martino"

3. We note that you have stated that you will provide repository information for your data at acceptance. Should your manuscript be accepted for publication, we will hold it until you provide the relevant accession numbers or DOIs necessary to access your data. If you wish to make changes to your Data Availability statement, please describe these changes in your cover letter and we will update your Data Availability statement to reflect the information you provide

Additional Editor Comments (if provided):

Please check the comments made by both reveiwers and submit you rebuttal not later than 17/10/2021. You are advice to disregard the comments left by Reviewer 1 in your revision wherever you find appropriate. 

Reviewers' comments:

Reviewer's Responses to Questions

**Comments to the Author**

1. Is the manuscript technically sound, and do the data support the conclusions?

Reviewer #1: Yes

Reviewer #2: Yes

2. Has the statistical analysis been performed appropriately and rigorously? 

Reviewer #1: N/A

Reviewer #2: N/A

3. Have the authors made all data underlying the findings in their manuscript fully available?

Reviewer #1: Yes

Reviewer #2: Yes

4. Is the manuscript presented in an intelligible fashion and written in standard English?

Reviewer #1: Yes

Reviewer #2: Yes

5. Review Comments to the Author

Reviewer #1: I will appreciate authors for choosing an important topic and been able to collect significant information. However, I am concerned regarding the writing style which is not what we call the 'academic style of writing'.

Methods are not written in a structured way. How is design iterative? Why definition of terms are given in methods? how come 32 interviews were conducted from 33 participants.

The way thematic analysis reported also requires reconsideration. Categorize similar codes into one theme or sub-theme. Too many overlapping sub-themes can be seen.

Discussion is good in sense of co-relating findings with other studies however, lacks message/solution from author on the issues highlighted.

Conclusion is too general to establish the significance of the current study.

Reviewer #2: Dear authors,

This paper is well written. Kindly check 'Children drop out from school to help financially support the family', perhaps the word 'help' can be dropped.

Please check all the references so that they are consistent with the journal's format.

6. PLOS authors have the option to publish the peer review history of their article (what does this mean?). If published, this will include your full peer review and any attached files.

Reviewer #1: No

Reviewer #2: No

---

## [Author Response · Author response to Decision Letter 0]

27 Sep 2021

Response to reviewers

Reviewer #1:

1. Methods are not written in a structured way. 

The methods section has been restructured with subsections for clarity. The sections are (1) Overview (2) Recruitment, (3) Data collection, (4) Data analysis and (3) Ethics.

We also have reorganised and edited the Methods section to improve the flow.

2. How is design iterative? 

We have rewritten this section removing the term ‘iterative’.

Line 93-97 in the methods section:

We used qualitative methods to explore experiences and challenges faced by marginalised non-citizen children in Malaysia and those supporting them, in accessing education during the COVID-19 pandemic. This study focuses on refugee and asylum seeker, migrant, stateless and undocumented children in Malaysia. International students and children of expatriates were excluded from this study. (Refer to S1 Text for the definition of terms)

3. Why are definition of terms given in methods? 

We acknowledge that the definition of key populations need not be included in the Methods section, since they have been sufficiently described in the Introduction section.

Instead, we moved the definition of terms to the appendix (S1 Text)

We also moved the inclusion and exclusion criteria to the initial introduction of the Methods. See Line 93-97 above.

4. How come 32 interviews were conducted from 33 participants.

Of these 32 in-depth interviews, we had 2 sessions where 2 participants from the same organisation were interviewed, while the rest were interviews had one participant only. We also conducted a second interview with one participant. So, in total 33 participants were interviewed.

We aimed to conduct individual interviews for this study. However, two organisations sent two representatives to participate in interviews. One participant was interviewed twice. 

Line 107 - 110 in the methods section:

Data collection was conducted from June 2020 to March 2021. We conducted 32 in-depth interviews, interviewing 33 individuals. Most interviews were conducted on an individual basis; however, two interviews were conducted with 2 participants from the same organisation. One participant was interviewed twice. 

5. The way thematic analysis reported also requires reconsideration. Categorize similar codes into one theme or sub-theme. Too many overlapping sub-themes can be seen

We have carefully read the results section and have agreed that organisation of main themes and subthemes should remain. However, we have substantially edited overlapping quotes and repetitious points. We hope that this would be able to convey the complexity of the situation with more clarity.

We itemise changes made here:

Page 13 – deleted Point

Parents were unable to afford to buy digital devices or mobile top-ups for data connectivity to support online learning during the pandemic.

Page 15 – deleted quote

“Students who did not have that support, they sort of disappeared. Although we registered them online. And we got them to make the connection, but they did not continue. Or they would do things like, you know, not [submit homework] on time. Maybe after two weeks, they will submit something.” EP-03 (August 2020)

Page 16 – deleted point

…to the lack of support and isolation forced by quarantine measures,…

… while being confined at home especially if the home environment was negative …

Page 16 – rewritten paragraph for clarity 

Refugees living in urban areas often live in cramped environments that pose a challenge to remote learning. Forced eviction further exacerbate the issue of the lack of space and privacy for effective learning.

Page 17 – deleted point

… or help set up remote learning…

Page 21- rewritten paragraph for clarity

Study participants informed that home classes were only conducted when case numbers were relatively low before October 2020. Educationists interviewed in early 2021, described physical visits to students' home as not possible at that time because of increased risk of COVID-19 transmission and the legal repercussions of not complying with movement restrictions. 

Page 22 – deleted quote

“In terms of education, the migrant communities, of course, are in a dire situation simply because there are a lot of organisations that are pulling out funding as well, right? So, there’s so much that needs to be done. You know you need to provide at least iPads or something for the children to learn online. But at the same time, there’s no funding for it. So, kids who are going to school and the school are being funded by the NGO school or faith-based schools…. I don’t think they are able to sustain because they are all going online [and] not all kids are able to go online. And the NGOs and do not have the capacity to provide for all these children.” RES-04 (Oct 2020)

Page 23 – rewritten paragraph

Interviewees shared that even though school fees were low, they were still a hurdle for marginalised populations. Non-citizen families were faced with financial difficulties due to job loss and immigration crackdowns against undocumented migrants. Educationists shared that some learning centres discounted tuition fees, as their main aim was to enable children to continue their education and not to generate profit. 

6. Discussion is good in sense of co-relating findings with other studies, however, lacks message/solution from author on the issues highlighted.

Thank you for your comment.

We feel that this paper highlights problems faced by children from marginalised communities in accessing education during the COVID-19 pandemic. These students have been excluded from mainstream education and are only able to access informal education through community and NGO run learning centers. This paper illustrates the struggles the community face and the efforts made by stakeholders to support the community and ensure educational access. There is no easy solution to this situation.

In this paper, we suggest that the lack of government oversight is in alternative learning centres and non-citizen children is an issue. We suggest that we should look at the example of neighboring Thailand that provides free education to non-citizen children, even though those schools have issues. 

Page 26 – additional paragraph for clarity 

Though there is no straightforward solution to ensuring rights to education access for vulnerable population, this study can serve as a useful input for future policy design to protect education rights for vulnerable children in the time of crisis. First, policy makers should be aware that the one-size-fit-all measure does not always lead to favourable outcomes. The education system for vulnerable children, particularly the non-citizens, in many places differ greatly from the mainstream education for the citizens. The abrupt change in policies (such as implanting an on-line learning or school closure), if done without supporting systems in place (in terms of budget, technology, finance and know-how), may cause longterm damage on the children (as a result of school dropout or poor learning outcomes) than the shortterm health damage caused by COVID-19. Preparedness plan for the education system for any future crises should seriously account for the diverse nature of education system for non-citizen children.

7. Conclusion is too general to establish the significance of the current study.

Thank you for your comment. We feel that the conclusion highlights that the pandemic has exacerbated pre-existing vulnerabilities faced by non-citizen communities. Non-citizen children are unable to enter government schools in Malaysia, thus are their welfare and education is looked after by NGOs, community, faith-based and international organisations.

In the conclusion, we urge the government to use the pandemic as an opportunity to be inclusive in provision of education and recommend that the government take an active role in ensuring non-citizen children are ensured the right to education.

 

Reviewer #2: 

1. Kindly check 'Children drop out from school to help financially support the family', perhaps the word 'help' can be dropped.

We made the necessary change to Table 2 dropping the word ‘help’

2. Please check all the references so that they are consistent with the journal's format.

We have changed the style on citation manager to the PLOS style and have checked the citations to ensure quality.

---

## [Decision Letter · Decision Letter 1]

21 Oct 2021

Education for non-citizen children in Malaysia during the COVID-19 pandemic: A qualitative study

PONE-D-21-23915R1

Dear Dr. Loganathan, 

We’re pleased to inform you that your manuscript has been judged scientifically suitable for publication and will be formally accepted for publication once it meets all outstanding technical requirements.

Kind regards,

Maryam Farooqui, Ph.D

Academic Editor

PLOS ONE

Reviewers' comments:

Reviewer's Responses to Questions

**Comments to the Author**

1. If the authors have adequately addressed your comments raised in a previous round of review and you feel that this manuscript is now acceptable for publication, you may indicate that here to bypass the “Comments to the Author” section, enter your conflict of interest statement in the “Confidential to Editor” section, and submit your "Accept" recommendation.

Reviewer #1: All comments have been addressed

2. Is the manuscript technically sound, and do the data support the conclusions?

Reviewer #1: Yes

3. Has the statistical analysis been performed appropriately and rigorously? 

Reviewer #1: N/A

4. Have the authors made all data underlying the findings in their manuscript fully available?

Reviewer #1: Yes

5. Is the manuscript presented in an intelligible fashion and written in standard English?

Reviewer #1: Yes

6. Review Comments to the Author

Reviewer #1: Dear Author.

Thank you very much for considering my suggestions.

Your manuscript is pretty much clear now and flows well.

7. PLOS authors have the option to publish the peer review history of their article (what does this mean?). If published, this will include your full peer review and any attached files.

Reviewer #1: No

---

## [Editor Report · Acceptance letter]

3 Nov 2021

PONE-D-21-23915R1 

Education for non-citizen children in Malaysia during the COVID-19 pandemic: A qualitative study 

Dear Dr. Loganathan:

I'm pleased to inform you that your manuscript has been deemed suitable for publication in PLOS ONE. Congratulations! Your manuscript is now with our production department. 

Kind regards, 

on behalf of

Dr. Maryam Farooqui 

Academic Editor

PLOS ONE